



# Use of expert elicitation to assign weights to climate and hydrological models in climate impact studies

Eva Sebok[1], Hans Jørgen Henriksen[1], Ernesto Pastén-Zapata[1,2], Peter Berg[3], Guillaume Thirel[4], Anthony Lemoine[4], Andrea Lira-Loarca[5], Christiana Photiadou[3,6], Rafael Pimentel[7,8], Paul Royer-Gaspard[4], Erik Kjellström[3], Jens Hesselbjerg Christensen[9,10,11], Jean Philippe Vidal[12], Philippe Lucas-Picher[13,14], Markus G. Donat[15,16], Giovanni Besio[17], María José Polo[7,8], Simon Stisen[1], Yvan Caballero [18], Ilias G. Pechlivanidis[3], Lars Troldborg[1], Jens Christian Refsgaard[1]

[1]Geological Survey of Denmark and Greenland, Copenhagen, Denmark
[2]University of Eastern Finland, Joensuu, Finland
[3]Swedish Meteorological and Hydrological Institute, Norrköping, Sweden
[4]Université Paris-Saclay, INRAE, HYCAR Research Unit, Antony, France
[5]University of Granada, Granada, Spain
[6]European Environment Agency, Copenhagen, Denmark
[7]Fluvial Dynamics and Hydrology Research Group, Andalusian Institute for Earth System Research (IISTA), University of Córdoba, Córdoba, Spain
[8]Department of Agronomy, Unit of Excellence María de Maeztu (DAUCO), University of Córdoba, Córdoba, Spain
[9]Physics of Ice, Climate and Earth, Niels Bohr Institute, University of Copenhagen, Copenhagen, Denmark
[10]NORCE Norwegian Research Centre, The Bjerknes Centre for Climate Research, Bergen, Norway
[11]Danish Meteorological Institute, Copenhagen, Denmark
[12]INRAE, UR RiverLy, Villeurbanne Cedex, France
[13]Groupe de Météorologie de Grande Échelle et Climat, Centre National de Recherches Météorologiques, Météo-France, Toulouse, France
[14]Département des sciences de la Terre et de l'atmosphère, Université du Québec à Montréal, Montréal, Québec, Canada
[15]Barcelona Supercomputing Center, Barcelona, Spain
[16]ICREA, Pg. Lluís Companys 23, Barcelona, Spain
[17]University of Genoa, Genoa, Italy
[18]BRGM, Univ. Montpellier, Montpellier, France

*Correspondence to*: Hans Jørgen Henriksen (hjh@geus.dk)

**Abstract.** Various methods are available for assessing uncertainties in climate impact studies. Among such methods, model weighting by expert elicitation is a practical way to provide a weighted ensemble of models for specific real-world impacts. The aim is to decrease the influence of improbable models in the results and easing the decision-making process. In this study both climate and hydrological models are analyzed and the result of a research experiment is presented using model weighting with the participation of 6 climate model experts and 6 hydrological model experts. For the experiment, seven climate models are a-priori selected from a larger Euro-CORDEX ensemble of climate models and three different hydrological models are chosen for each of the three European river basins. The model weighting is based on qualitative evaluation by the experts for each of the selected models based on a training material that describes the overall model structure and literature about climate models and the performance of hydrological models for the present period. The expert elicitation process follows a three-stage approach, with two individual elicitations of probabilities and a final group



consensus, where the experts are separated into two different community groups: a climate and a hydrological modeller
group. The dialogue reveals that under the conditions of the study, most climate modellers prefer the equal weighting of
ensemble members, whereas hydrological impact modellers in general are more open for assigning weights to different
models in a multi model ensemble, based on model performance and model structure. Climate experts are more open to
exclude models, if obviously flawed, than to put weights on selected models in a relatively small ensemble. The study shows
that expert elicitation can be an efficient way to assign weights to different hydrological models, and thereby reduce the
uncertainty in climate impact. However, for the climate model ensemble, comprising seven models, the elicitation in the
format of this study could only reestablish a uniform weight between climate models.

## 1 Introduction

Uncertainty of future climate projections is a key aspect in any impact assessment, such as hydrological impacts (Kiesel et
al., 2020; Krysanova et al. 2017). Hydrological impact modelling often involves regional downscaling of global scale
simulations and bias adjustment of the multiple driving variables for multiple ensemble members of multiple global and
regional climate models, as well as multiple greenhouse gas emission scenarios (Pechlivanidis et al., 2017; Samaniego et al.,
2017). With added uncertainties in each step in this chain (also known as a cascade) (Mitchell and Hulme, 1999; Wilby and
Dessai, 2010), the number of simulations can quickly become overwhelming and especially, the uncertainty can become
inflated (Madsen et al., 2017). The end results will contain a mixture of sampled uncertainties stemming from core
climatological processes, and methodological and statistical influences on the results.

The large computational burden and the huge projection uncertainties are difficult to cope with for practitioners. Therefore,
decision makers in the water sector using climate services have increasingly demanded a user-friendly, tailored, high-
resolution climate service (Vaughan and Dessai, 2014; Olsson et al., 2016; Jacobs and Street, 2020) that preferably
incorporates a reduction of computational burden and projection uncertainty (Dessai et al., 2018; Krysanova et al. 2017).
Information on the confidence of the climate change projection and impacts result is often not sufficiently transparent for
end-users (Schmitt and Well, 2016). This is an important barrier for the implementation of adaptation options (Klein and
Juhola, 2014; Brasseur and Gallardo, 2016) and constrains the efficiency of climate services.

There are aspects of climate where some members of a model ensemble can be proven more trustworthy than others, such as
the simulation of key atmospheric circulation patterns at global or local scale, and specific features of particular importance
for a case study. It can be argued that these ensemble members should be promoted above others, or given larger weights in
an ensemble statistic, when evaluating the climate change projections. If some climate models have very low trustworthiness
and in practice can be discarded this reduces the computational requirement. If the models that are discarded or given low
weight have projections furthest away from the ensemble mean, potentially as a consequence of missing process
descriptions, model weighting may in addition result in a reduced uncertainty. In this respect there are different traditions in
the climate and hydrological modelling communities.





The climate modelling community often prefers using a large ensemble of climate models and ensemble model weighting is a controversial issue. Model democracy (Knutti, 2010) is a well-established term in the climate modelling community referring to the widespread assumption that each individual model is of equal value and when combining simulations to estimate the mean and variance of quantities of interest, they should be unweighted (Haughton et al., 2015). The claim for
model democracy is supported by the argument that the value of weighting climate models has not been clearly demonstrated (Christensen et al. 2010; Christensen et al. 2019; Matte et al. 2019; Clark et al. 2016; Pechlivanidis et al. 2017; Samaniego et al. 2017), or that model weighting simply adds another level of uncertainty (Christensen et al., 2010). At the same time, in recent years, there has been a significant effort on sub-selecting models from the large ensemble of models based on different frameworks (diversity, information content, model performance, climate change signal etc.). Here, the argument is
that model democracy has not been useful for impact modelling with the purpose of adaptation (see investigations in Kiesel et al. (2020); Pechlivanidis et al. (2018), Wilcke and Bärring (2016), Knutti et al. (2013)). Another practical reason for selecting a smaller sub-set of representative ensemble members from the larger ensemble is that impact modelling can be computationally and methodologically intensive in case a large number of models has to be applied (Kiesel et al 2020). The hydrological modelling community, on the other hand, typically uses a small ensemble of hydrological models (e.g., Giuntoli
et al., 2015; Karlsson et al., 2016; Broderick et al., 2016; Hattermann et al., 2017) and model weighting using Bayesian Model Averaging or other methods is quite common and non-controversial (Neumann, 2003; Seifert et al., 2012).

Climate projections of precipitation (Collins, 2017) and more generally hydrological variables are subject to large uncertainty. This has been the motivation for utilizing an expert judgement methodology to assess the impact of model uncertainty. Expert judgment techniques have previously been used to estimate climate sensitivity (Morgan and Keith,
1995), future sea level rise (Bamber and Aspinall, 2013; Horton et al, 2020), credibility of regional climate simulations (Mearns et al. 2017), and tipping points in the climate system (Kriegler et al., 2009). One such technique, called expert elicitation (EE), is frequently used to quantify uncertainties, in decision making or in cases with scarce or unobtainable empirical data (Bonano et al., 1989; Curiel-Esparza et al., 2014). However, the application of EE to regional climate change has largely been undocumented, underspecified or incipient, with a few exceptions (Mearns et al., 2017). Given the large
uncertainties in projecting regional and local climate change, Thompson et al. (2016) have argued that subjective expert judgment should play a central role in the provision of such information to support adaptation planning and decision making. Ideally, this kind of expert judgement should be carried out in a strictly defined group of experts dealing with the topics addressed by the impact model.

There are different ways to sub-select more trustworthy members from a large multi model ensemble of climate and impact
model projections e.g. so-called emergent constraints or observational constraints (Hall et al. 2018). An alternative approach to looking at model quality for the historical climate focus on to sub-selecting ensemble members spanning the uncertainty range related to the future climate change signal (Wilcke and Bärring 2016). Contrary to these quantitative methods, EE is a more qualitative technique that assesses the trustworthiness of single members based on the subjective knowledge of experts (Ye et al., 2008; Sebok et al., 2016). One possible way to describe the uncertainties of climate models and hydrological



impact models is by model weighting where the experts assign probabilities to the different models, which is used to weight the different members of an ensemble of models (Morim et al. 2019; Risbey and O'Kane, 2011; Chen et al., 2017). In our context, EE uses expert judgement and dialogue (intersubjectivity) to assign weights within an ensemble of climate and hydrological models for specific real-world applications. Basically, such weights are (inter)subjective and prone to have inherent biases (Tversky and Kahneman, 1974), thus they must be derived following a transparent protocol and process

(Morgan and Keith, 1995; Ye et al., 2008; Bamber and Aspinal, 2013; Sebok et al., 2016; Morim et al. 2019).

Building narratives, for example, in regional climate change through EE is one option (Hazeleger et al., 2015; Stevens et al., 2016; Zappa and Shephard 2017; Dessai et al., 2018). Thompson et al. (2016) argue that this is needed when providing climate service information to support adaptation planning and decision making. Further, several users also need to perform in-depth exploration of each step of the chain and for both frequent and more rare events.

The aim of the present study is to test the EE method to provide weighted ensembles of climate and hydrological models for specific real-world cases. As a result, this approach will weight and rank those models from the a-priori selected ensemble models, which have the highest perceived probability of reliably projecting climate change and hydrological impacts, for clearly stated catchment specific issues. This investigation has the following specific objectives:

- To investigate EE and expert judgement to provide weighted ensembles of climate and hydrological models for
specific real-world impacts.
- To analyze the individual- and group-elicitation of probabilities in model selection, and the dialogue between experts from the two communities (climate and hydrology) and their impact on the individual probabilities.
- To identify lessons learned regarding the format of the expert elicitation and to identify alternative designs for overcoming weaknesses discovered in the EE.

In Section 2, the case studies and climate and hydrological models are described. Section 3 describes the methods for the EE, including selection of experts, planning, training and aggregation of results. Section 4 describes the results from the two groups (climate and hydrological modelers). Section 5 discusses the EE results along with the aggregated uncertainties from the individual and group elicitations and discusses virtual versus in-person workshop. Finally, issues of our EE are discussed before concluding in Section 6.

**2 Case studies and climate and hydrological models**

**2.1 Case studies**

Five case studies distributed across different hydroclimatic zones in Europe were used (Fig. 1). These case studies are located in Sweden, Denmark, France and two sites in Spain (Fig. 1). Only the Danish, French and Spanish Guadalfeo river catchments' (cases #2, #3 and #4) were included in the hydrological model assessment. The cases have different aims and,

therefore, require different information from climate services. Additionally, the sites have contrasting climate and physical characteristics (Table 1.). For instance, the observed annual precipitation trends are positive for the Swedish and Danish





cases and negative for the remaining sites. The experts that participated in the elicitation were given a training material document describing the most important characteristics of each case study (see supplementary information).

### 2.2 Climate models

The climate model ensemble consisted of different regional climate model (RCM) combinations from the Euro-CORDEX initiative (Jacob et al., 2014) available at a 12.5 km x 12.5 km resolution. The RCMs are driven by global climate models (GCMs) from CMIP5 (Coupled Model Intercomparison Project Phase 5, Taylor et al., 2012). The model combinations selected for the analysis included all models that fulfilled the following criteria (at the time of extraction from ESGF in May 2019):

• Scenarios driven by Representative Concentration Pathways (RCP) 2.6, 4.5 and 8.5 (van Vuuren et al., 2011).
   • Daily outputs of precipitation, 2-m (mean, maximum and minimum) air temperature, 10-m wind speed and sea level pressure.
   • Available coverage for simulation period from at least 1971 to 2099.

Even though the above criteria might not be relevant for the EE, it integrates different requirements that climate models
should fulfill to develop a comparable impact assessment. The seven selected climate model combinations (Table 2) were used for the analysis at all sites. The limited number of climate models also conforms to the limits of the elicitation method as previous studies found that experts are expected to make less reliable judgements when ranking more than 7 items (Miller, 1956; Meyer and Booker, 2001). Detailed performance of the climate models compared to the observations and info3rmation on teleconnection patterns/atmospheric variability patterns was given to the experts before the workshop as training material
(see supplementary material).

### 2.2 Hydrological models

Three different hydrological models were used for each of the French, Danish and the Spanish Guadalfeo river case studies (Table 3). The selected hydrological models are frequently employed in each of the sites to assess the impacts of climate change on hydrology. Here we argue that an assessment of the models, which are commonly used in each site, is more
relevant than assessing the same ensemble of models at all sites. Consequently, the hydrological models presented here are a mixture of distributed, physically-based, semi-distributed and lumped conceptual models, depending on the site under assessment. The hydrological models used in each site are briefly presented in Table 3**Error! Reference source not found.**. An extensive description of the models and their performance in each of the sites were given to the experts before the elicitation as training material (see supplementary material).



## 3 Expert elicitation experimental setup

Expert elicitation is a formal method of uncertainty assessment often used in studies where due to the sparse or unobtainable empirical data the experience and subjective opinion of experts is used as additional input (Krayer von Krauss et al., 2004). In this study the elicitation comprised a climate and hydrological modelling perspective with a similar aim to find the models with the highest probability of reliably projecting climate change and climate change impacts within a model ensemble. Initially, the EE was planned to take place in March 2020 in the form of a joint in-person workshop, where climate and hydrological modelling experts could have participated both in plenary and topical sessions and discussions. However, due to the outbreak of the COVID-19 pandemic, a virtual setting with two separate workshops was adopted even though expert elicitation is traditionally conducted in the form of in-person sessions. The elicitation took place in form of virtual workshops on 25-26 May 2020 for climate modelling experts and for hydrological modelling experts on 3-4 June. The separation of climate and hydrological modelers at the workshops hindered discussions that were planned to take place between the two groups of experts. On the other hand, moving the elicitation to a virtual platform gave an excellent opportunity to explore how virtual elicitation could work in the future.

Even though the elicitation was moved from in-person to a virtual platform, the training material and the elicitation structure remained as originally planned. The elicitation was centered around a questionnaire which the experts were asked to fill in during three consecutive elicitation steps. The first two steps included individual evaluations while the last step comprised a group elicitation where experts were asked to reach consensus about the questionnaires (Fig. 2).

### 3.1 Selection of experts

During the planning of the elicitation study, 18 selected experts were invited to contribute to the elicitation and altogether 12 experts had accepted to participate in the study by December 2019. The two virtual workshops were planned with the participation of these 12 experts; 6 hydrology and 6 climate experts.

The role of the experts was to provide the knowledge necessary to assess members of climate or hydrological model ensembles through individual assessment and group discussions. These experts were selected based on recommendations from the partner institutes of the research project. As a requirement, the experts were previously not affiliated with the elicitation experiment, but some degree of familiarity to the geographical area of at least one of the case studies was expected. Correspondingly, for the hydrological modelling experts, experience with at least one of the hydrological models was regarded as an additional selection criterion. After the second individual round of elicitation, one hydrological modelling expert decided to leave the study, thus only the five remaining experts participated in the group elicitation. The probabilities assigned by this expert are included in the results of the study.



### 3.2 Formulating elicitation questions

The elicitation both for the climate and hydrological models had the same approach taking place in three consecutive elicitation steps including both individual and group elicitation (Fig. 2). During each step, the experts were asked to fill in a questionnaire (see supplementary material). A separate questionnaire was developed for the climate and hydrological modelling experts following the same principles. The questionnaire was composed of two separate blocks of progressively quantitative questions.

The first block was aimed at making the experts conscious about the elicited climate and hydrological models by asking for their assessment on modelling concepts, structures and assumptions that can influence the models' ability to predict climate or hydrological processes under future conditions and thus influence the probability assigned to the models. This block included questions where the experts first had to make a qualitative assessment of the elicited models, then were asked to rank the models according to their capabilities in predicting future climate and specific hydrological (or related) processes in

the study areas. Answers to this block of questions were only elicited during the first two individual steps of the elicitation, but not during the group elicitation (Fig. 2).

Eliciting answers to the second block of questions was the main purpose of elicitation, where experts were expected to assign probabilities to the climate and hydrological models for each case study. It was possible to assign zero probability to models, while the total probability assigned to models had to equal 1.

This progressively quantitative approach of first making qualitative assessment, then ranking (block 1) and finally assigning probabilities (block 2) was chosen to ease the experts into making decisions on model probabilities, as studies have shown that it comes more naturally to experts to make qualitative assessment or ranking than assign probabilities (Goossens and Cooke, 2001). The purpose of the iterative structure was two-fold, firstly it gave an opportunity to the experts to revise their opinions, secondly it also enabled consistency checks of the individual experts' answers.

### 3.3 Planning the elicitation


Step 1 of the elicitation was an individual elicitation, where experts were asked to fill in a questionnaire (Fig. 2). It was expected that experts would use their previous experience, intuitive knowledge about the hydrological/climate models and also rely on the training material provided by the workshop organizers. The completed questionnaires were supposed to be returned a week before the workshop. Nine experts returned the first questionnaire before the proposed date, while three

experts delivered them just before the online workshops two months later, thus some experts had a fresher memory of the questionnaire when filling it out again during the workshop.

In Step 2, experts were asked again to individually fill in the questionnaires at the end of the first day of the virtual workshop (Fig. 2). It was assumed that the presentation of the case studies, climate and hydrological models and the discussions of the anonymous results of the first individual round of elicitation during the workshop would potentially clarify issues, provide





the experts with new ideas introduced by fellow experts and give the experts an opportunity to re-evaluate their initial opinion. The results of this elicitation round could also reflect on how the experts influence each other.

In Step 3, as a last step of the elicitation process (Fig. 2), the climate and hydrological modelling experts were asked to participate in a moderated discussion with the aim of reaching consensus on the probabilities assigned for each model for the specific case studies. The discussions were moderated by scientists of the AquaClew project, with one moderator for the

climate group and three moderators with specific knowledge about modelling in each hydrological model case study for the hydrological modeller group. Participants of the research project were also listening to the conversation helping the moderator with comments or suggestions for questions in the background unnoticed by the participating experts. Prior to the group elicitation, experts were also shown the anonymous results of the second individual elicitation round, followed by a short discussion. Between each round of elicitation, the anonymous results of the previous elicitation round were shown to

the experts and experts were given the opportunity to discuss and comment on the results (Fig. 2).

### 3.4 Training of the experts on the case studies and elicitation

A training material describing the concept and the aim of EE, case study catchments, the climate models and the hydrological models was sent to the experts four months prior to the workshops (see supplementary material). In the training material, the scientists of the AquaClew project described all case studies and both the climate and hydrological models in a

similar manner with the same indicators of model performance, which are based on comparing their simulation skill to observational datasets. Even though such comparison is common to assess the simulation skill of Euro-CORDEX climate models (e.g., Kotlarski et al., 2014; Casanueva et al., 2016), it is acknowledged that there is a degree of uncertainty coming from the observation datasets (e.g., Herrera et al., 2019; Kotlarski et al., 2017). It was requested that experts familiarize themselves with this training material and if suitable, include it in their assessment during the elicitation.

During the first day of the workshop, the case studies were again presented to the experts, who were also reminded about the concept of EE and the biases that could influence their judgement during the elicitation (Fig. 2). The most common biases that were expected to occur during the elicitation (overconfidence, anchoring, availability and motivational bias) were also demonstrated. It was emphasized that the method relies on the experts' subjective assessment based on prior knowledge and experience and the general impressions of the training material. Questions could also be skipped in case the experts were not

comfortable answering.

### 3.5 Aggregation of results

As the aim of the elicitation was to assign probabilities for both climate and hydrological models to assess which ones are deemed to be most reliable in describing climate change and climate change impacts, only the second block of questions eliciting probabilities will be presented in detail. Both for the climate and hydrological models, the probabilities were elicited

three times, twice individually leading to a mathematical aggregation, and at last as a group elicitation involving behavioral aggregation (Fig. 2) where the group of experts had to reach consensus.



For the first two rounds of elicitation, the individual assessment of experts was used to calculate the 50th percentile of the probability distribution for each model following the process described by Ayyub (2001). For this, the six probabilities given by the experts for a specific model were first ranked in decreasing order, then the arithmetical mean of the third and fourth highest probability was used to calculate the 50th percentile of the probability distribution for the specific model (Eq. (1)):

$$Q = (X_3 + X_4)/2 \hspace{4cm} (1)$$

where the arithmetical mean of the third ($X_3$) and fourth ($X_4$) highest probability of the ranked expert probabilities is used to calculate the 50th percentile of the probability distribution (Q) for each model. As the 50th percentile of the probability distribution was calculated for each model independently, the sum of 50th percentiles within the model ensembles will not necessarily be equal to 1.

Results from the group elicitation were obtained by discussion, where the six climate and five hydrological modelling experts managed to reach consensus on the second day of the virtual workshops (Fig. 2). The group elicitation comprises more than assigning probabilities, which is the direct output, it is also an expert inquiry or dialogue, which eventually can be used for bringing in new ideas or identifying new issues for inquiry.

## 4 Results

As the first block of qualitative questions was only aimed at preparing the experts to make quantitative decisions on model probabilities, only the probability results for the second block of quantitative questions will be presented (see questionnaire in the supplementary material).

### 4.1 Aggregated probabilities – Hydrological model results

As for the hydrological modelling group, both the number of case studies and the number of models in the hydrological ensembles was lower, results will first be shown for this group. The individual results from step 1 and 2 of the elicitation were aggregated mathematically, while for step 3 the group discussion led to behavioral aggregation (Fig. 2).

Probabilities for the three models in the French case study had little spread compared to the other case studies (Fig. 3a). For the first individual elicitation, the GR4J model was assigned the highest 50th percentile of probabilities with 0.35. The experts assigned a slightly lower value of 0.33 to GR6J, while TOPMO got 0.3. During the second round of individual elicitation, all models were assigned the same 50th percentile probability of 0.33. The third group elicitation made a slight differentiation of the models, again the GR4J and GR6J models both got a consensus probability of 0.35 and 0.35, respectively, while TOPMO was assigned a probability of 0.3 (Fig. 3a, Table 4).

For the first round of individual elicitation in the Danish case study, the 50th percentile of probabilities was the highest for the Two-Layer model with 0.40, slightly lower for the Richards' equation with 0.35, and considerably lower for the Gravity flow model with 0.20 (Fig. 3b). The ranking remained the same for the second round of individual elicitation with a slight change in probabilities as four out of six experts revised their probability values. Thus, the Two-Layer model had the highest





50th percentile of 0.38, while the probabilities of the Richards' equation model remained approximately the same and probabilities for the Gravity flow model increased to 0.25. In the third round of elicitation, the group of hydrological

modelling experts reached a consensus about the model probabilities assigning a probability of 0.38 to the Two-Layer model, 0.35 for the Richards' equation model and 0.27 for the Gravity flow model (Fig. 3b, Table 4).

In the Spanish case, the first round of individual elicitation resulted in a 50th percentile of 0.45 for the SWAT model, while the WiMMed model was assigned a value of 0.42, the HYPE model was clearly deemed the least probable model (Fig. 3c). In the second round of individual elicitation the assigned probabilities for the HYPE model slightly increased to 0.25, while

the experts differentiated more clearly the SWAT and WiMMed models, assigning the highest probabilities of 0.40 to the WiMMed model. This distribution of probabilities was maintained also in the third round of elicitation where the group assigned a consensus probability of 0.45, 0.30 and 0.25 to the WiMMed, SWAT and HYPE models, respectively (Fig. 3c).

The experts of the hydrological modeller group gave variable probabilities to all models of the ensemble although for the French case the probabilities had a small spread (Fig. 3a). None of the experts assigned zero probability to models. In

summary, expert judgement about hydrological models stayed rather stable along the multiple steps of the elicitation. Although discussions between experts led to small adjustments in the probabilities of a few models (e.g. HYPE and SWAT in the Spanish case study), the overall model ranking did not change through the elicitation steps Fig. 3). The experts also reached a consensus rather easily as a group in the last phase. The willingness to assign variable probabilities and the ease with which the experts reached consensus in the last step of elicitation could also be attributed to the small number of models

in the ensemble or because they found it easy to develop a constructive consensus process. For instance, for the Spanish case, the expert who had more experience developing studies in the area gave a detailed explanation on why and how he/she assigned the probabilities to the models. Even though the expert explained and shared strong motives, the other experts were also involved in the discussion, exchanging comments and finally reaching a consensus that gave probabilities close (but not the same) to the ones assigned by the expert that detailed his/her selection process.

**4.2 Mathematically aggregated probabilities – Climate model results**

For the Danish case study, when calculating the 50th percentile of the probabilities given by experts for the seven climate models, the highest probability of 0.21 was assigned to the EC-EARTH-CCLM GCM-RCM modelling chain in the first round, and of 0.19 to the EC-EARTH-RACMO model combination in the second individual round of elicitation (Fig. 4a). The lowest probability of both the first and second round of individual elicitation was assigned to the models EC-EARTH-

RCA4 and MPI-ESM RCA4 with values of 0.09 and 0.08, respectively (Table 5).

For the French case study, the EC-EARTH-CCLM GCM-RCM combination had the highest 50th percentile probability, respectively 0.20 and 0.18, for both rounds of individual elicitation, whereas in the second round of elicitation the MPI-ESM-REMO models was also assigned a similarly high calculated probability value (Fig. 4b). The MPI-ESM-RCA4 received the lowest probability of 0.10 in the first individual round of elicitation, while the EC-EARTH-RACMO obtained a

similarly low probability in the second round of elicitation (Table 5).



For the Spanish Sierra Nevada case study of water resource allocation, the 50th percentile of probabilities was the highest for the EC-EARTH-CCLM and HadGEM-RACMO models in the first round with a value of 0.14 (Fig. 4c). While in the second round, the highest probability was 0.16 for the EC-EARTH-CCLM models. The MPI-ESM-RCA4 model combination received the lowest probability of 0.08 for both individual elicitation rounds (Table 5).

For the Granada case study of fluvial and coastal interactions in Spain, the MPI-ESM-RCA4 combination received the lowest probabilities of 0.08-0.09 for both individual elicitation rounds (Table 5). The 50th percentile was the highest for EC-EARTH-RACMO models in the first round with a probability of 0.15. In the second individual round both the EC-EARTH-CCLM and EC-EARTH-RACMO combinations were ranked the highest with probabilities of 0.16 for 50th percentile of probabilities (Fig. 4d).

For the Swedish case study, the 50th percentile of probabilities was the highest for the HadGEM-RACMO combination with 0.18 and 0.19 in the first and second elicitation round, respectively (Fig. 4e). The EC-EARTH-RCA4 combination was assigned the lowest probabilities of 0.08-0.1 in both elicitation rounds (Table 5).

During the first two rounds of elicitation, Expert 3 gave equal probability to each of the climate models irrespective of the case study, while another expert also assigned zero probability to several climate models and a very high probability to others or, in the two Spanish case studies, zero to all the listed climate models. The remaining four experts gave varying

probabilities to the climate models depending on the case studies (Fig. 4). It is assumed that the six experts had two different approaches to assign probabilities to climate models. Experts 3, 5 and 6 assigned all model combinationss of the ensemble, irrespective of the case study, the same probability values or values only slightly different from equal probability, while the other three experts had a wider range of probability values. This distinction between the approaches is most apparent for the

two Spanish case studies (Fig. 4c, d) and could probably related to the potential influence of snow and wind in these two case studies as mentioned by several experts during the group discussion.

### 4.3 Change of probabilities between the two individual elicitation rounds

Individual expert opinions were elicited in two consecutive steps (Fig. 2), thus giving an opportunity to experts to revise their opinion. This revision in individual opinions could be due to a change in opinion as new ideas were introduced by other

experts or also to a clearer understanding of elicitation concepts, notions or elicitation questions. This revision of subjective opinions however is deemed necessary in the iterative elicitation approach.

Considering the climate modelers, two or three out of six experts changed their opinion between the first and second round of individual elicitation depending on the case study. Experts 1 and 2 made the largest changes in their probability assessment between the elicitation rounds (Fig. 4). The largest change in probability was recorded for the French case study

by Expert 2, while for the Spanish case study of fluvial and coastal interactions the change between the first and second elicitation rounds was minimal. For the hydrological modellers, three or four experts out of six altered their assigned probabilities also depending on the case study (Fig. 3). The expert leaving the study did not wish to modify the assigned probabilities based on the input of the workshop. The largest change in assigned probabilities was observed for Expert 4 in



the Spanish case study. The smallest change between elicitation rounds was observed in the French case study, where only
Expert 3 made large changes in the probability distribution (Fig. 3a).

Compared to the climate modeller group, the hydrological modeling experts were more willing to change their opinion between the elicitation rounds and also issued larger changes in probabilities. This could be due to the lower number of models in the ensemble or to the fact that differences between the assigned probabilities between the experts were smaller.

### 4.4 Behavioural aggregation of probabilities – reaching consensus

In the last step of elicitation (Fig. 2), the groups were asked to reach a consensus about the probabilities of the ensemble members for each case study, thus expert opinion was expressed through a behavioural aggregation. For the climate modeller group, in the group elicitation step the experts reached a consensus where all climate models within the ensemble had the same probability of 0.14 for each case study (Fig. 4), reflecting that it is not possible to differentiate the reliability of model ensemble members to project climate change based on the available information on their ability to simulate past climate.
This result agrees with the individual opinion of Expert 3, who maintained an equal-probability approach throughout the entire elicitation study. Thus, the opinion of this expert clearly influenced the group decision, while Expert 5 and 6 also showed a similar approach assigning only slightly different probabilities to the ensemble members in the first two elicitation rounds (Fig. 4). At the same time, instead of ranking the members of the model ensemble, the experts raised the idea of potentially excluding some models which were deemed to be less likely. Due to the influence of snow in the French and one
of the Spanish Guadalfeo river catchment case studies, the experts had a lengthy discussion about potentially downranking some climate models due to temperature biases, but finally stated that the observational basis was too weak to support the rejection of any of the models.

Our assumption is that this change in results between the second round of individual elicitation and group elicitation can mostly be attributed to the principles of model democracy and to the fact that weighting of ensemble members is a
controversial issue in scientific literature with recent research on the methods of optimal weighting of local to regional scale climate models. This also encompasses that climate experts were not comfortable with the EE methodology as a potential way of assigning weights to individual climate models. This could be due to a general lack of confidence due to the subjectivity of the method, an unfortunate phrasing of the aim and questions of elicitation, a lack of time or as raised by some of the experts the lack of relevant information in the training material. Some experts were concerned that the model
performance information provided in the training material was not sufficiently relevant to judge the model performance for the specific case studies and therefore did not serve as a robust basis for downranking individual models.

In the hydrological modellers group, the experts assigned variable probabilities to the models, keeping the ranking of the individual elicitation rounds (Fig. 3). It was observed that some experts had a geographical expertise about the case study areas or some of the models in the ensemble (as was required as an expert selection criterion). During the group discussion,
the experts were candid about their expertise and the group took advantage of this specialized knowledge when reaching consensus about probability values. Thus, while some experts were more involved in group decisions than others for specific





case studies, there was not one expert who dominated the whole discussion. The opposite was also observed, experts who did not have previous experience with a member of the model ensemble clearly accepted the group opinion without trying to influence it. This kind of influence or lack of arguments was mostly observed in relation to the models of the ensemble and

not the geographical experience with the case study areas. As a general conclusion, hydrological modelling experts were more willing to acknowledge that all models may not have an equal probability to predict changes in a future climate, while also trying to accommodate EE as a potential method to assign probabilities to hydrological models.

The group consensus results were also compared to the results of the second elicitation round. For the hydrological modelling experts, the 50th percentile of the probability distribution had similar results to the group consensus, with the

largest difference of 0.065 in probability for the SWAT model for the Spanish case study (Fig. 3). The ranking of model probabilities did not change between the second individual elicitation round and the group consensus, however the relative difference in the probability of the models was revised (Table 4). Due to this slight change and the discussion which allowed a better understanding of challenges in relation to the models and geographical areas, it is assumed that the group discussion was a necessary part of the elicitation despite the minor changes in probabilities. In the case of the climate modelling group,

it was only during the last step of group elicitation where the collective opinion of climate modellers transpired leading to equal probabilities to all members of the climate model ensemble (Table 5). Therefore, it can be concluded that the elicitation could not have been successfully made in less steps as it would not have truthfully reflected the opinion of climate experts.

During the discussion of the elicitation results, the hydrological modelling experts expressed their doubts about assigning

probabilities to the models as the models' capability to project future hydrological processes depends on the parameterization and the purpose of the case study. Here, it was questioned if the experts are assigning similar probabilities, because the models have a similar capability of predicting future hydrological processes, or because they cannot distinguish between the models. Such could be the case for the Danish case study where the same hydrological modelling software was used with a different representation of the unsaturated zone and evapotranspiration.

In both the hydrological and climate modeller groups, peer-pressure was present as all experts wished to reach a consensus by the end of the allocated timeframe. This was especially observed at the hydrological modelling group, where experts assigned variable probabilities to the models, thus having a lengthier discussion for each case study. Even though the climate modelling experts first agreed to assign equal probability to all climate models for each case study, they nevertheless had discussions if some members of the climate model ensemble could have slightly differing probability values. A factor

frequently mentioned, most likely leading to variable probabilities, was the representation of snow by the climate model, when relevant for the case study.





## 5 Discussion

### 5.1 Expert elicitation as a tool for uncertainty assessment

EE is frequently used to quantify uncertainties in decision making in case of scarce or unobtainable empirical data (Bonano
et al., 1989; Curiel-Esparza et al., 2014). In the context of this study, EE is used to assign weights to an ensemble of climate and hydrological models to identify more trustworthy models. The main advantage of using EE probabilities instead of "objective tests" is for the obvious reason that no data exist for the unknown future. Of course, by using historical data, it may be possible to perform numerical tests for present conditions or also past changes (Refsgaard et al., 2014; Kiesel et al., 2020). In a climate change context however, these tests are not entirely reliable, due to the large uncertainties associated with
future climate projections. Moreover, expert elicitation as conducted in this study as a dialogue between experts in a group setting, incorporates more than just the subjective, individual opinion of different experts in the uncertainty assessment. Through an individual elicitation, experts can work with unpredictability and incomplete knowledge, but in a group elicitation and dialogue, experts can in addition exchange and deal with multiple knowledge frames and modify their opinion on the basis of common knowledge (Brugnach et al., 2009). Hereby, EE can potentially provide additional information
compared to quantitative approaches.

Another advantage of expert elicitation could be that it is computationally not demanding, as it is based on previously acquired knowledge, such as already existing models. Thus, it could require less time to obtain some results about the capability of models to reliably predict climate change than to run several models in an ensemble, calculating comparable statistics and in case of hydrological models, also calibrating the models. At the same time, due to its subjectivity, expert
elicitation is also prone to biases. In this study, anchoring was observed as some experts did not wish to revise their initial opinion about probabilities. This bias is however considered of small importance for the hydrological modelling part as the final results were reached during the group discussion which is independent of the experts' initial opinion. For the climate modeller group, the final results agree with the principles of model democracy assigning equal weight to all models of the ensemble, reflected by three experts who maintained this opinion throughout the elicitation, thus potentially influencing the
outcome of the group discussion.

Our study suggests that expert elicitation can be a suitable methodology to assign probabilities to hydrological models applied for climate change impact assessments as these probabilities are variable and robust across the expert panel and across the different elicitation steps and discussions (Table 4). This finding is in line with Ye et al. (2008) who used expert elicitation to assign probabilities to recharge models used to simulate regional flow systems. The results from using expert
elicitation to discriminate between climate models were less conclusive. As during the group discussion, experts were not willing to rank or even exclude climate models of the ensemble, expert elicitation in the form of our study is not suitable to select a subset of GCM-RCM climate change projections for the future based on their performance in reproducing aspects of the historical climate.



### 5.2 Different outcomes from the climate and hydrological model groups

Even though the elicitation questionnaire had the same approach and structure both for the climate and hydrological modeller groups, there is a distinct difference in the reception of the elicitation methodology and the responses to the questionnaire between the climate and hydrological modeller groups. With the exception of the expert who left the study after the first day of the workshop, hydrological modellers were more ready to assign variable probabilities to different hydrological models acknowledging that some models have a higher probability to correctly project climate change when

using the same climate model input. At the same time, climate modelling experts agreed that based on the information available at the elicitation all climate models should have equal probability of accurately projecting the future climate, thus, an ensemble could reach a wide range of potential climate projections. However, in the climate modelling community there is currently a significant effort on sub-selecting or weighting models based on suitable quantitative information (Wilcke and Bärring 2016; Donat et al., 2018; Hall et al., 2019) and for impact studies with adaptation purposes such approaches is

widely accepted among modellers (Krysanova et al., 2018).

This difference between the approach of the climate and hydrological modelling group could have several reasons. There are many hydrological model codes in use and research groups routinely create, develop, or modify model codes according to their site-specific modelling purposes (Pechlivanidis et al., 2011). Thus, some model codes are only used by a small fraction of the hydrological modelling community. As an example, in the hydrological model ensemble for the Danish case study, all

models had the same basis, only the conceptualization of the unsaturated zone was different. In contrast, as the climate models are applied at regional to global scales with increasing computational demand, there are fewer models in use often sharing common parametrizations. These climate models also frequently require collaboration of research groups and generally applied on much coarser spatial scales.

Hydrological models are typically used to make quantitative predictions. Thus, they are calibrated by optimizing model

parameter values and only accepted as suitable if they reasonably match observation data. Classical hydrological modelling studies use one hydrological model code to make predictions, but in recent decades there is an increasing tendency for model intercomparison studies (Dankers et al., 2014; Krysanova et al., 2017; Christierson et al. 2012; Chauveau et al. 2013; Giuntoli et al. 2015; Vidal et al. 2016; Warszawksi et al., 2014) attempting to evaluate model uncertainties or which model codes are the most suitable for making such predictions. This also means that hydrological modellers are used to the idea of

evaluating model results or even assigning weights to models. Climate models often stem from short-term forecast models, however, with a shift in emphasis from an initial value problem to a boundary condition problem, which also shifts focus from calibration to the predicted surface variables to the main energy and water budget and main climate processes (Hourdin et al., 2017). We note that the initial value problem is again introduced in decadal predictions (Boer et al., 2016; Meehl et al., 2021), however, here we focus on the applications to long term climate change. Therefore, the kinds of uncertainties

typically explored in a climate model ensemble relate to the climate processes, and not so much to bias in surface parameters used in hydrology. A climate model may have merit in describing the changing climate, even if it displays bias in the





historical period. Still, several studies have investigated different means of reducing the climate model ensemble by their inter-dependencies (e.g. Knutti et al., 2013), their bias in specific variables (e.g. Christensen et al., 2010), or by the information content in individual ensemble members (e.g. Pechlivanidis et al., 2018). Studies that attempt sub-selection of

climate models generally find a strong dependence on the variables used, location, season and future scenario (Wilcke and Bärring, 2016, Pechlivanidis et al., 2018).

One of the key findings from the EE experiment was that the climate modellers were reluctant to discriminate or assign probabilities for the different climate models. Instead, they agreed on assigning equal weights. When taking a closer look on the discussion in the climate model group, it becomes clear that this finding is conditional to the information about case

studies and hydrological models presented in the training material. Experts lacked a more comprehensive material potentially including model evaluation data on simulated changes and variabilities supplying available information given on mean states and biases. Additional need for information such as on representative of surface energy balance, large-scale atmospheric circulation, representation of cloudiness was mentioned. In this elicitation study evaluating climate models proved to be a task that requires a broader view of model specificities than the information relative to the cases studies provided in the

training material. For this reason, expecting experts of the climate groups to assess different weights to the models was possibly inadequate.

Another feedback from the dialogue in the climate modeller group was that seven climate models is not enough, and that more information / a matrix is required for providing probabilities (anonymous quotes):

*Individually they [the climate models] only make sense when the rest of them is also there… doing a probability*
*assessment for these, there is no objective way to do that, the only objective way to go ahead is to say…ok, we have a set of seven experiments, in order to make sense of any one of them, all the other ones have to be there as well…in the sense that we don't have a qualifier to disregard any of the models, we have to accept all of them.*

*This exercise reminds me about …the ENSEMBLES project [note: Hewitt et al., 2005]. If we really want to provide a weight, we need to have a matrix… we don't have a matrix we only have our gut feeling…I don't think at this*
*point we can really provide numbers… I found really small differences between different models…*

The climate modellers supposedly would have been much more willing to give specific advice based on expert elicitation in case they would have had many more models to work from. Whether they would recommend removing some of the worst models from an ensemble is a possibility. Instead, we searched another way forward by asking the climate modellers for their recommendation if they should select a subset of models. Below answers from three experts are quoted about selecting

only four models from an ensemble of seven:

*There are quite many people working on that. How to select from different ensembles…span some kind of uncertainty ranges in different dimensions…not just look whether models are realistic or not, but also looking at some kind of span, looking at ranges…*





*My advice would be that you don't use models. You just tell your gut feeling about the result, and that is as good as any model, because four models will not provide robust information...I would not bet my property on four models...You need a much more qualified information...*

*It is not a proper way to portray uncertainty...*

We believe that with a different design for the expert elicitation that provides climate experts with more information on hydrological impact simulations and aims at clarifying how to best select a few RCM projections that best sample the spread

of climate projections, the elicitation could potentially have resulted in down-ranking some model combinations of the ensemble. The conclusion of our study is that in this expert elicitation setting climate modellers deem each climate model to have equal probability and are extremely unlikely to exclude or down-rank any climate model unless they are without doubt inappropriate. Instead of model democracy, impact studies in practice are recommended to use a range of subset of available climate models to reduce computational time and effort (Wilcke and Bärring, 2016). For sub-selecting models, more

objective methods than the present elicitation study are preferred.

## 6 Conclusion

As part of the AquaClew research project an expert elicitation experiment with a group of six climate model experts and another group of six hydrological model experts was carried out in May-June 2020 in a virtual setting. The aim of the elicitation was to assign weights to members of climate and hydrological model ensembles following a strict, multi-step

protocol including two steps of individual and one final step of group elicitation with the same structure but separate sessions for climate and hydrological modelers.

The experiment resulted in a group consensus among the climate modellers that all models should have an equal probability (similar weight) as it was not possible to discriminate between single climate models, while also maintaining the importance of using as many climate models as possible in order to cover the full uncertainty space in climate model projection. The

hydrological modellers also reached consensus after the group elicitation. However, the agreement here did result in different probabilities for the three hydrological models in each of the three case studies. For the hydrological modellers, the final group consensus results did not differ significantly from the results of the second individual elicitation round. Based on the results of this study, expert elicitation can be an efficient way to assign weights to hydrological models, while for climate models, the elicitation in the format of this study only re-established model democracy.

Due to the Covid-19 pandemic we were forced to shift the setting from an in-person to a virtual workshop. We conclude that the design and protocol used for the expert elicitation was satisfactory also in the virtual setting, where the new virtual platforms provide an alternative to in-person meetings. However, the virtual setting was more demanding for the moderators to ensure the equal engagement of each participating expert and for all participants due to practical issues of working from home in the very early stages of the pandemic.



**Author contribution**

ES, HJH, EPZ, JCR, PB and CP designed the study, ES, HJH, EPZ, PB, GT, AL, ALL, CP, RP and PRG assembled the scientific material, organized the expert elicitation workshop. EK, JHC, JPV, PLP, MGD, GB, MJP, SS, YC, IGP and LTR acted as experts in the elicitation workshop. ES analyzed the results, ES, HJH, EPZ, PB, GT, AL, ALL, CP, RP, PRG and JCR discussed the results, ES, HJH and EPZ prepared the manuscript and all authors contributed on the review of the

manuscript.

**Acknowledgements**

This work was funded by the project AquaClew, which is part of ERA4CS, an ERA-NET initiated by JPI Climate and funded by FORMAS (SE), DLR (DE), BMWFW (AT), IFD (DK), MINECO (ES), ANR (FR) with co-funding by the European Commission [Grant 69046]. The contribution of P. Lucas-Picher was supported by the French National Research

Agency under the future investment program ANR-18-MPGA-0005. Rafael Pimentel acknowledges funding by the Modality 5.2 of the Programa Propio-2018 of the University of Cordoba and the Juan de la Cierva Incorporación program of the Ministry of Science and Innovation (IJC2018-038093-I). Rafael Pimentel and María J. Polo are members of DAUCO, Unit of Excellence ref. CEX2019-000968-M, with financial support from the Spanish Ministry of Science and Innovation, the Spanish State Research Agency, through the Severo Ochoa and María de Maeztu Program for Centers and Units of

Excellence in R&D.

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



| Case study location | Sweden | Denmark | France | Guadalfeo, Spain | Granada, Spain |
|---|---|---|---|---|---|
| Aim of the assessment | Climate change impacts on the biodiversity | Climate change impacts on the foundation of agricultural production | Climate change impacts on hydropower production and water management for other uses | Climate change impacts on the allocation of water for tourism, agriculture and energy | Changes in physical processes such as sea waves, fluvial discharges and sediment transport |
| Area (km²) | 10,000 | 1,124 | 3,580 | 530,5 | |
| Precipitation (mm/yr) | 741 | 1,003 | 1,055 | 745 | 460 - 630 |
| Temperature (Celsius) | 5.6 | 8.8 | 3.2 | 12.5 | - |
| Wind speed (m/s) | - | - | - | - | 18-22 |
| Wind waves (km) | - | - | - | - | 200 - 300 |

**Table 1. Aim of the different study cases along with their physical and observed climate characteristics**



| GCM | RCM | Realization (Member #) |
|---|---|---|
| EC-EARTH | RACMO22E | 12 |
| EC-EARTH | CCLM4-8-17 | 12 |
| EC-EARTH | RCA4 | 12 |
| HadGEM2-ES | RCA4 | 1 |
| HadGEM2-ES | RACMO22E | 1 |
| MPI-ESM-LR | RCA4 | 1 |
| MPI-ESM-LR | REMO2009 | 1 and 2 |

**Table 2 Climate model ensemble used for expert elicitation. The realization (member #) column denotes the version of initial conditions.**



| Site | Models | Characteristics | References |
|------|--------|-----------------|-----------|
| Denmark | MIKE-SHE: Two-layer | • Physically-based, integrated and fully distributed model at a resolution of 250m x 250m<br>• The model divides the unsaturated zone into a root zone and a zone below the root zone<br>• Vertical flow depends on the storage of the unsaturated zone, ignoring the delay in the flow<br>• Actual evapotranspiration is a function of the potential evapotranspiration and the soil moisture content | Abbott et al., 1986; Graham and Butts, 2005; Yan and Smith, 1994 |
| | MIKE-SHE: Gravity flow | • Physically-based, integrated and fully distributed model at a resolution of 250m x 250m<br>• Based on the continuity equation and Darcy's law<br>• Vertical flow only depends on the force of gravity<br>• Actual evapotranspiration is a function of vegetation and the soil moisture content | Abbott et al., 1986; Graham and Butts, 2005; Kristensen and Jensen, 1975 |
| | MIKE-SHE: Richards' equation | • Physically-based, integrated and fully distributed model at a resolution of 250m x 250m<br>• Based on the continuity equation and Darcy's law<br>• Vertical flow depends on the soil moisture retention and hydraulic conductivity<br>• Actual evapotranspiration is a function of vegetation and the soil moisture content | Abbott et al., 1986; Graham and Butts, 2005; Kristensen and Jensen, 1975 |
| France | GR4J | • Lumped conceptual model with four parameters<br>• Water balance controlled by actual evapotranspiration and groundwater<br>• Snowmelt is simulated using the two-parameter CemaNeige model | Perrin et al., 2003; Valéry et al., 2014 |
| | GR6J | • Lumped conceptual model with six parameters<br>• Modified version of GR4J that allows a change of the direction of the exchange and adds a conceptual store<br>• Snowmelt is simulated using the two-parameter CemaNeige model | Perrin, 2000; Le Moine, 2008; Valéry et al., 2014 |
| | TOPMO | • Lumped conceptual model adapted from TOPMODEL with seven parameters<br>• Water balance controlled by evaporation from interception and groundwater reservoirs<br>• Simulates runoff combining heterogeneous time response flows from exponential and quadratic routing stores<br>• Snowmelt is simulated using the two-parameter CemaNeige model | Michel et al., 2003; Beven and Kirvy, 1979; Valéry et al., 2014 |
| Spain (Guadalfeo River) | HYPE | • Semi-distributed conceptual model<br>• Infiltration is estimated using a water table discrimination model<br>• Evaporation is estimated using the modified Hargreaves-Samani method<br>• Snowmelt is simulated using three decay factors related to temperature, radiation and fractional snow cover | Lindström et al., 2010; Samuelsson et al., 2011 |
| | SWAT | • Semi-distributed conceptual model<br>• Infiltration is estimated using the Green and Ampt method for a single soil layer<br>• Evapotranspiration is estimated using the Penman-Monteith method<br>• Snowmelt is simulated using the degree-day method | Arnold et al., 1988; Green and Ampt,1911; Penman 1948; Monteith et al., 1964 |
| | WiMMed | • Distributed physically-based model<br>• Infiltration is estimated using the Green and Ampt method for a two-soil layer<br>• Evapotranspiration is estimated using the Penman-Monteith method<br>• Snowmelt is simulated using a punctual energy and mass balance extended to cell scale using depletion curves | Polo et al., 2009; Aguilar et al., 2011, Herrero et al., 2009, Pimentel et al., 2017, |

**Table 3 Hydrological models used in each of the study sites**



| Danish case study | Step 1 50th percentile | Step 2 50th percentile | Group consensus |
|---|---|---|---|
| Two-Layer | 0.4 | 0.38 | 0.38 |
| Gravity flow | 0.2 | 0.25 | 0.27 |
| Richards' equation | 0.35 | 0.35 | 0.35 |
| French case study | Step 1 50th percentile | Step 2 50th percentile | Group consensus |
| GR4J | 0.35 | 0.33 | 0.35 |
| GR6J | 0.33 | 0.33 | 0.35 |
| TOPMO | 0.30 | 0.33 | 0.3 |
| Spanish case study | Step 1 50th percentile | Step 2 50th percentile | Group consensus |
| HYPE | 0.18 | 0.35 | 0.25 |
| SWAT | 0.45 | 0.37 | 0.30 |
| WiMMed | 0.42 | 0.40 | 0.45 |

**Table 4 Assessed probabilities by hydrological model experts**




| Danish case study | Step 1 50th percentile | Step 2 50th percentile | Group consensus |
|---|---|---|---|
| 1-EC-EARTH-CCLM | 0.21 | 0.16 | 0.14 |
| 2-EC-EARTH-RACMO | 0.13 | 0.19 | 0.14 |
| 3-EC-EARTH-RCA4 | 0.09 | 0.08 | 0.14 |
| 4-HadGEM-RACMO | 0.18 | 0.17 | 0.14 |
| 5-HadGEM-RCA4 | 0.14 | 0.14 | 0.14 |
| 6-MPI-ESM RCA4 | 0.09 | 0.08 | 0.14 |
| 7-MPI-ESM REMO | 0.14 | 0.14 | 0.14 |
| French case study | Step 1 50th percentile | Step 2 50th percentile | Group consensus |
| 1-EC-EARTH-CCLM | 0.20 | 0.18 | 0.14 |
| 2-EC-EARTH-RACMO | 0.12 | 0.10 | 0.14 |
| 3-EC-EARTH-RCA4 | 0.11 | 0.13 | 0.14 |
| 4-HadGEM-RACMO | 0.15 | 0.15 | 0.14 |
| 5-HadGEM-RCA4 | 0.12 | 0.14 | 0.14 |
| 6-MPI-ESM RCA4 | 0.10 | 0.12 | 0.14 |
| 7-MPI-ESM REMO | 0.15 | 0.18 | 0.14 |
| Spanish case study 1 SN | Step 1 50th percentile | Step 2 50th percentile | Group consensus |
| 1-EC-EARTH-CCLM | 0.14 | 0.16 | 0.14 |
| 2-EC-EARTH-RACMO | 0.13 | 0.15 | 0.14 |
| 3-EC-EARTH-RCA4 | 0.12 | 0.14 | 0.14 |
| 4-HadGEM-RACMO | 0.14 | 0.14 | 0.14 |
| 5-HadGEM-RCA4 | 0.13 | 0.14 | 0.14 |
| 6-MPI-ESM RCA4 | 0.08 | 0.08 | 0.14 |
| 7-MPI-ESM REMO | 0.09 | 0.09 | 0.14 |
| Spanish case study 2 G | Step 1 50th percentile | Step 2 50th percentile | Group consensus |
| 1-EC-EARTH-CCLM | 0.14 | 0.16 | 0.14 |
| 2-EC-EARTH-RACMO | 0.15 | 0.16 | 0.14 |
| 3-EC-EARTH-RCA4 | 0.11 | 0.10 | 0.14 |
| 4-HadGEM-RACMO | 0.15 | 0.15 | 0.14 |
| 5-HadGEM-RCA4 | 0.14 | 0.10 | 0.14 |
| 6-MPI-ESM RCA4 | 0.09 | 0.09 | 0.14 |
| 7-MPI-ESM REMO | 0.11 | 0.12 | 0.14 |
| Swedish case study | Step 1 50th percentile | Step 2 50th percentile | Group consensus |





| | | | |
|---|---|---|---|
| 1-EC-EARTH-CCLM | 0.14 | 0.18 | 0.14 |
| 2-EC-EARTH-RACMO | 0.14 | 0.15 | 0.14 |
| 3-EC-EARTH-RCA4 | 0.10 | 0.09 | 0.14 |
| 4-HadGEM-RACMO | 0.18 | 0.19 | 0.14 |
| 5-HadGEM-RCA4 | 0.14 | 0.16 | 0.14 |
| 6-MPI-ESM RCA4 | 0.11 | 0.12 | 0.14 |
| 7-MPI-ESM REMO | 0.14 | 0.15 | 0.14 |

**Table 5 Assessed probabilities by climate model experts**

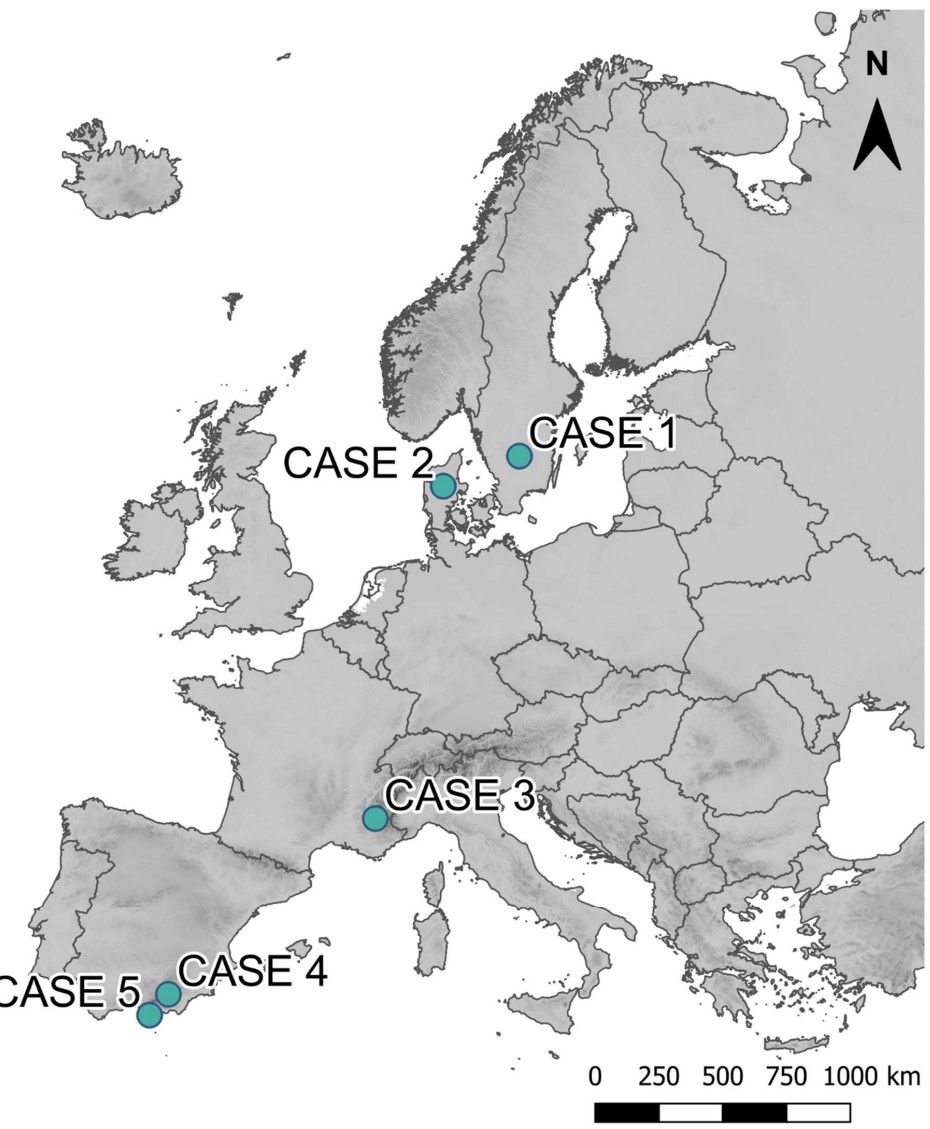

**Fig. 1 The location of case studies within Europe. Only cases 2, 3 and 4 are included in the hydrological modelling part of the**
**expert elicitation**





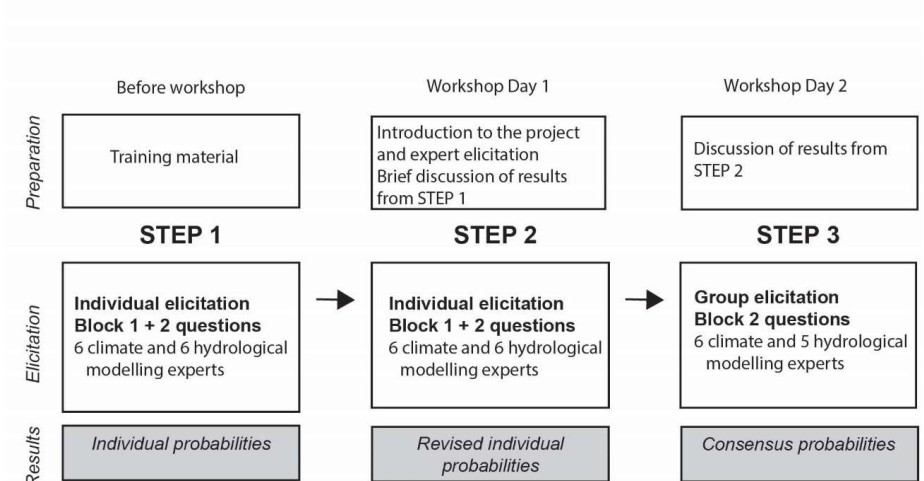

**Fig. 2 Flow chart of the expert elicitation process**





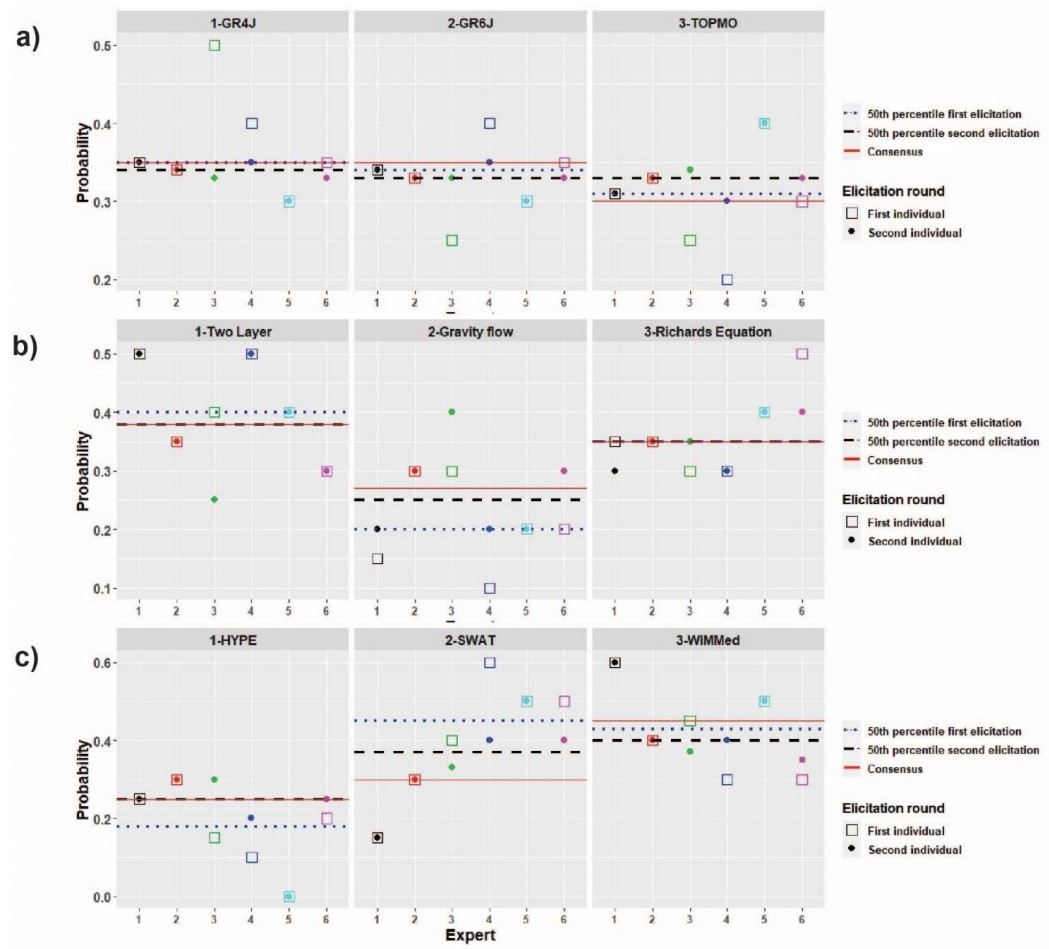


**Fig. 3 Probabilities assigned by the experts for the three alternative hydrological model structures in the French (a), Danish (b) and Spanish(c) case studies after first and second individual rounds and group elicitation**





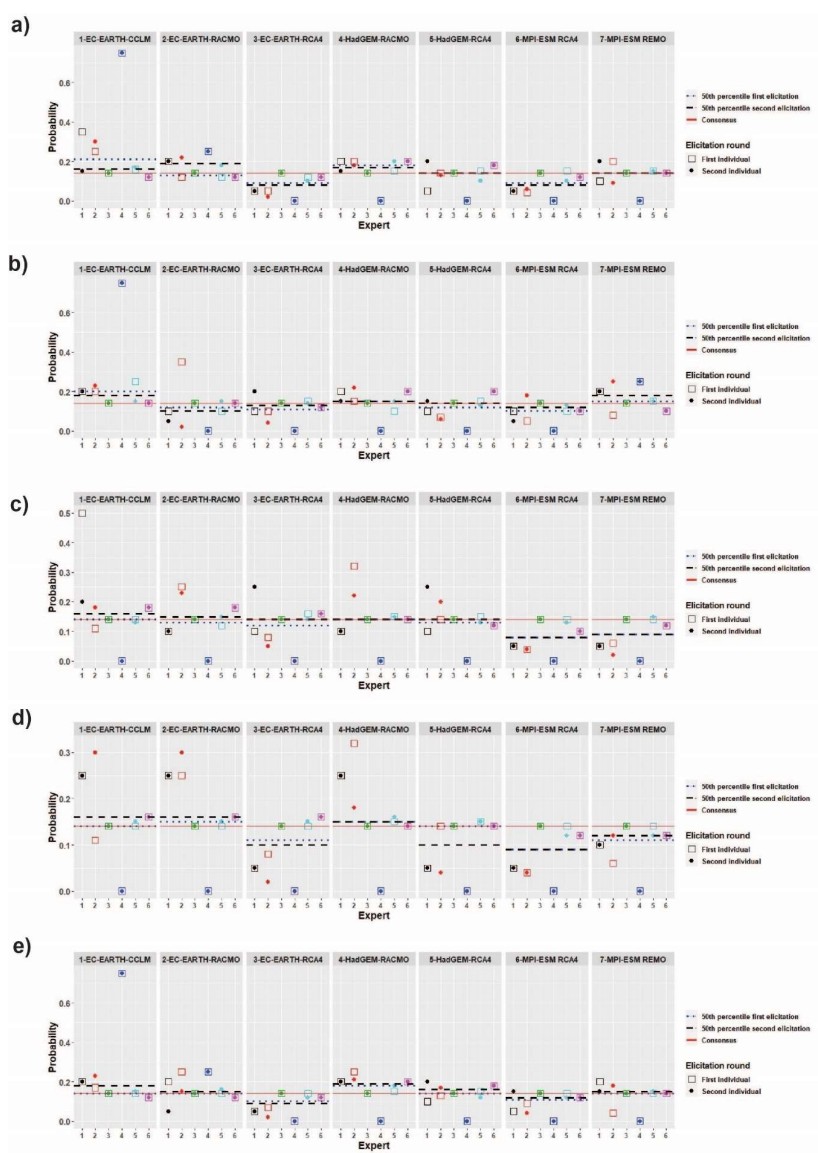

**Fig. 4 Probabilities assigned by the experts to the seven selected climate models in the Danish (a), French (b), Spanish Sierra Nevada (c), Spanish Granada (d) and Swedish (e) case studies after the first and second individual elicitation and for the group elicitation**