# Peer review of "Use of expert elicitation to assign weights to climate and hydrological models in climate impact studies"

_Hydrology and Earth System Sciences, 2021_

## Author Comment (AC1)

**Response to reviewers' comments to the manuscript: 'Use of expert elicitation to assign weights to climate and hydrological models in climate impact studies' (manuscript number hess-2021-597)**

First of all the authors would like to thank the two anonymous reviewers for the useful and encouraging comments! Based on the suggestions we believe that we managed to address all concerns of the reviewers and generally improve the clarity of the manuscript.

Please note that the references to page, line and figure numbers in the corrected manuscript refer to the revised manuscript submitted together with this response.

In the following, we listed how we responded to the review comments:

**Response to Reviewer #1**

Unfortunately, despite an admirable effort by the authors to produce a robust paper, it is a fatally flawed approach to assess impacts.

Here are several papers that discuss this issue:

Burgess et al: 2020: IPCC baseline scenarios have over-projected CO2 emissions and economic growth Environmental Research Letters 16 (1), 014016

Pielke Jr R. and J Ritchie: 2021: Distorting the view of our climate future: The misuse and abuse of climate pathways and scenarios R Pielke Jr, J Ritchie Energy Research & Social Science 72, 101890 Pielke Jr R. and J Ritchie: 2021: How Climate Scenarios Lost Touch With Reality R Pielke Jr., J Ritchie Issues in Science and Technology, 74-83

Pielke Jr et al 2021: Most plausible 2005-2040 emissions scenarios project less than 2.5 degrees C of warming by 2100 R Pielke Jr, MG Burgess, J Ritchie SocArXiv

Thanks for the comments with suggestions for references. We don't attempt to provide a methodology for selecting or weighing of climate models from different emission scenarios. Rather, our approach is limited to weighting of different GCM/RCMs climate models within a given RCP ensemble (e.g. RCP2.6, RCP4.5 or RCP8.5 from the AR5 Cordex dataset).

Designing and using ensembles for regional climate change assessments to take uncertainty into account is acknowledged in the recent literature (see WG1 / IPCC AR6 report section 10.3.4.4. page 10-62 to 10-63). The AR6/WGI report mentions and highlights that (see page 10-63 line 1-4): "A number of emerging methodologies aim at optimizing the ensembles available by weighting the simulation results according to a number of criteria relevant at the regional scale that aim at obtaining more realistic estimates of the uncertainty (Sanderson et al., 2015; Brunner et al. 2020)". This is what our paper attempts to evaluate.

We have made these conditions in our paper more clear in our introduction with two new references to Burgess et al. 2020 and Pielke and Ritchie 2021: "In this paper we focus on available climate scenarios from CMIP5 without looking into uncertainties of different RCP scenarios. We do not address the uncertainties related to how future socio economic growth impact the suite of IPCC emission scenarios, where some scientists argue that IPCC RCP8.5 high emission (baseline) scenario over-project future CO2 emissions (Burgess et al. 2020; Pielke and Richie, 2021)" lines (63-66).

The more robust way to assess risk is the contextual approach proposed by:

Füssel, H.-M. (2007), Vulnerability: A generally applicable conceptual framework for climate change research, Global Environ. Change, 17, 155–167.

O'Brien, K. L., S. Eriksen, L. Nygaard, and A. Schjolden (2007), Why different interpretations of vulnerability matter in climate change discourses, Clim. Policy, 7(1), 73–88.

Applications of this approach can be found in:

Hossain, F., J. Arnold, E. Beighley, C. Brown, S. Burian, J. Chen, S. Madadgar, A. Mitra, D. Niyogi, R.A. Pielke Sr., V. Tidwell, and D. Wegner, 2015: Local-to-regional landscape drivers of extreme weather and climate: Implications for water infrastructure resilience. J. Hydrol. Eng., 10.1061/(ASCE)HE.1943-5584.0001210, 02515002.

Pielke, Sr. R.A., J. Adegoke, F. Hossain, and D. Niyogi, 2021: Environmental and social risks to biodiversity and ecosystem health – A bottom-up, resource-focused assessment framework. Earth, 2, 440–456. https://doi.org/10.3390/earth2030026

Thank you for this comment. We don't disagree about the need for incorporating vulnerability and exposure in robust assessment of climate change impacts and risks. But since our main focus is on how climate models predict changes in precipitation, dry spells, evapotranspiration etc. and how hydrological models simulate future changes in hydrology (soil moisture, groundwater level, discharge etc.) in order to inform risk assessment (see also IPCC AR6 WG2 chapter 4), we believe that the probability of climate or hydrological models in an ensemble of models is a first necessary step to address as part of risk assessment.

We have added a clarification on our focus about informing risk assessment: "Instead we focus on the uncertainties related to a selected ensemble of climate model inputs (precipitation, dry spell, temperature, evapotranspiration etc.) and hydrological models (snow, soil moisture, groundwater depth, discharge etc.) used for informing risk assessment specific catchments in Europe." lines (66-68).

These uses of scenarios have become a cottage industry, but are poor science in my view.

If the authors still disagree, they need to quantitatively show in hindcast runs that the models can skillfully predict changes in regional climate statistics that matter to the hydrological impacts they are assessing. Reanalyses (of changes in regional climate statistics) are the baseline to compare with the models not between models.

This statement from their paper summarizies the inadequacy of the study: "The experiment resulted in a group consensus among the climate modellers that all models should have an equal probability (similar weight) as it was not possible to discriminate between single climate models, while also maintaining the importance of using as many climate models as possible in order to cover the full uncertainty space in climate model projection."

The reviewer points at a relevant question about the accuracy of climate models. This has been analyzed in several studies (Wang et al., 2017; Stryhal and Huth, 2019; Kotlarski et al., 2014; Casanueva et al., 2016). However, this is not the issue that is dealt with in our manuscript. The important question is not how well climate models can reproduce past climate, but how good they are in projecting future changes. We accept that the adequacy of models to simulate past events under today's greenhouse gas forcing is relevant and should be part of the analysis, and this information was available to the climate experts that contributed, when they assessed the

**ability of the different climate models to make projections of climate changes. However, analysis of the performance of climate models in simulating past climate, is outside the objectives of our manuscript.**

The uncertainty of the model results does NOT bracket the real world uncertainty. These types of studies are misleading policymakers.

We agree that the ensemble of climate models does not bracket the real world uncertainty and this has not been claimed in our manuscript, which does not attempt to address the issue raised by the reviewer in this comment. Indeed, the climate modellers strongly suggested adding more models to the study (line 533-536) to better span the uncertainty space, albeit still with the limitations of the models.

I checked "reconsider after major revisions", rather than recommending "rejection" since the authors' methodoloy of accepting the climate model results as having demonstrated skill at multidecadal regional climate change statistics is applied throughout the impacts communities. The authors need to objectively respond to the view that the approach they are using is not scientific robust.

Here we do not agree with the reviewer and believe it is scientifically robust to put weight on climate models and hydrological models in a selected ensemble, because it increases the reliability of the projection and gives insight on the probability of different models (we are not addressing specific adaptation measures in our paper, only how to increase the reliability of the projection among models selected for risk assessment).

**References:**

- Casanueva, A., Kotlarski, S., Herrera, S. *et al.* Daily precipitation statistics in a EURO-CORDEX RCM ensemble: added value of raw and bias-corrected high-resolution simulations. *Clim Dyn* 47, 719–737 (2016). https://doi.org/10.1007/s00382-015-2865-x
- Kotlarski, Sven, et al. (2014). Regional climate modeling on European scales: a joint standard evaluation of the EURO-CORDEX RCM ensemble. *Geoscientific Model Development* 7: 1297-1333.
- Stryhal, J., & Huth, R. (2019). Classifications of winter atmospheric circulation patterns: validation of CMIP5 GCMs over Europe and the North Atlantic. *Climate Dynamics*, 52(5-6), 3575-3598.
- Wang, X., Li, J., Sun, C., & Liu, T. (2017). NAO and its relationship with the Northern Hemisphere mean surface temperature in CMIP5 simulations. *Journal of Geophysical Research: Atmospheres*, 122(8), 4202-4227.

**Response to Reviewer #2**

**Summary:**

The paper uses five case study locations in Europe to test an expert elicitation approach to weighting climate and hydrological models. The study uses a structured expert elicitation approach, involving three stages of individual elicitations and consensus building, supported by some initial training material. The study finds that hydrologists are more willing than climate modellers to articulate model weights, with climate modellers preferring model democracy in the absence of further investigation. The shift in approach from in-person to online, due to the covid-19 pandemic, affected the approach followed away from typical elicitation processes.

Overall, I found the paper interesting, well written and clearly structured. The previous review comments seemed to have missed the point that this paper is focused on testing a methodology – i.e., expert elicitation to support model weighting – rather than providing robust scenarios of future climate impacts for the case study locations. As such, I do not share their concerns. Moreover, precisely because "all models are wrong, some are useful", I think the community should welcome efforts to more rigorously include expert judgement in providing actionable information, as relying purely on outputs from models (which we know to be flawed) risks over-confidence in uncertainty estimates.

That being said, the study isn't as conclusive as I'd hoped. The finding that climate scientists continued to support model democracy – whilst interesting in the context explored – is not surprising. I suspect the finding would have been different if those involved had seen more model validation results – i.e. ability of the simulations to capture observed atmospheric circulation and trends, relevant to precipitation in the locations studied. It is also a shame that the climate modellers and hydrological modellers weren't part of the same expert group as originally envisaged, as this may have yielded some more nuanced views and outcomes for model weighting.

Overall, I think the paper is worthy of publication. It will help advance the use of expert elicitation methods in the climate and hydrology community. I hope my comments and suggestions below help in improving the paper prior to publication.

**Thank you for the encouraging and constructive remarks, we believe that they helped to improve the clarity of our manuscript and raised some interesting issues to be studied further.**

**Specific comments:**

Line 94: "with a few exceptions (Mearns et al., 2017)" – add to this the recent study by Grainger et al. 2022 – see references. It would be interesting for the authors to comment on this study and how the methods followed compare, noting that there are very few studies in this space.

Thank you for bringing this interesting paper to our attention. As there are very few elicitation studies in this subject area it definitely merits mentioning in our manuscript. We however would omit further discussion about the study as both in methodology and topic it is significantly different from our elicitation. The study included a quantitative elicitation where experts had to provide a box-whiskers diagram on specific values related to climate change, while the only comparable element of the elicitation was the ranking of uncertainties. This ranking however did not raise similar issues as the ranking/weighting of climate and hydrological models. On the other hand both studies agree that it is useful to not solely rely on models, but on multiple knowledge types potentially through expert elicitation

**Reference added and an additional reference included: 'Hereby, EE can potentially provide additional information and knowledge that is absent from modelling approaches (Grainget et al., 2022). (lines 448-449)**

Line 102: Also worth citing McSweeney et al. 2015 – see references. This highly cited paper demonstrates a method for first excluding implausible models following model evaluation, and then spanning the uncertainty range of the remaining plausible models.

**Reference added**

Line 150: Table 2 shows 7 GCM-RCM model combinations, but the final combination includes two realisations meaning there are 8 simulations considered. Why is this not discussed in the paper, and why were two realisations selected for this model combination?

The table shows all GCM-RCM combinations that conformed to the elicitation design. However, the skill of the model with two realizations is very similar among both realizations, thus it was treated as one during the actual elicitation. This has been clarified: "*The selection resulted in a total of eight downloaded model experiments from the ESGF node, because the combinations of MPI-M-LR and REMO included two different realizations of the GCM, this was considered as a single model in the expert elicitation, but included separately in the training material to show the impact of natural variability in the one GCM-RCM combination.*" (lines 157-160)

In order to avoid further confusion, one of the realizations was deleted from the table.

Line 153: Typo "info3rmation"

**Corrected**

Line 162: Typo / technical error in referencing Table 3 in the pdf – check.

**Corrected**

Line 177: It is mentioned that the move to virtual elicitation gave an opportunity to explore how this worked. However, the paper doesn't provide much analysis here. I would encourage a short paragraph in the discussion section to reflect more on the pros and cons of this change in approach.

A paragraph discussing the virtual elicitation was added to the discussion: 'The virtual elicitation workshop was logistically simple, more cost-efficient and environmentally friendly as experts did not need to travel to the same venue. Thus, depending on the form of elicitation, more experts could potentially be involved in a virtual elicitation study. As opposed to a virtual elicitation, physical meetings are more natural, and experts are more engaged in informal dialogues and the elicitation. It is assumed that during an in-person meeting, experts are more inclined to ask questions, thus it is easier to anticipate, recognize and clear misunderstandings or handling the conflicts (such as in this study, one of the experts quitting the workshop). The practical experience with the virtual workshop was that it required more effort from the moderators to make sure that all the experts are involved in the discussion. Similarly, due to the lack of non-verbal communication, reaching consensus was also more time-consuming as experts had to be

individually asked to provide their opinion or agreement. Facilitators need to be more prepared to moderate meetings and engage experts. This necessitates increased awareness and also potentially, such as in this project, the involvement of background personnel who also monitored the discussion helping the moderator with comments from the background. Despite all the differences, based on our study it can be concluded that next to the traditional in-person meetings and workshops, the expert elicitation method can also be transferred to a virtual meeting.' lines (460-472)

Line 187: How were the 18 experts determined by the "partner institutes of the research project"? Were there any explicit or implicit considerations - e.g., PhD in a relevant topic, papers published, involvement in CORDEX?

In some elicitation studies experts are indeed selected or even weighted based on objective criteria, such as years of relevant research experience, involvement in research projects, number of publications etc. In our study however, the selection of 18 experts was purely based on personal, subjective considerations as described in the manuscript: 'As a requirement, the experts were previously not affiliated with the elicitation experiment, but some degree of familiarity to the geographical area of at least one of the case studies was expected. Correspondingly, for the hydrological modelling experts, experience with at least one of the hydrological models was regarded as an additional selection criterion' (lines 200-203).

In practice it meant that the partner institutes had previous working experience with some experts also working in the research area, but not related to this research project. This geographical familiarity and experience with the models was assumed to be crucial when assessing climate or hydrological models.

**No changes made in the manuscript.**

Line 192: It would help demystify things for the reader if you briefly explain why one expert decided to leave the study.

The following section was added to results: 'The expert leaving the study did not wish to modify the assigned probabilities based on the input of the workshop. One of the main reasons mentioned for quitting the workshop was that the workshop and the discussions did not contribute with any other relevant information than the previously studied training material, thus not being helpful in reaching a more sound assessment of the studied hydrological models. Coupled with the logistical difficulties of participating from home in a virtual workshop resulted in the decision of leaving the study.' lines (366-371)

Line 432: I agree obtaining results from models requires time, but the elicitation approach followed is also very time consuming and incurs a cost. I'm not sure saving time is a strong justification for following an elicitation approach.

We agree with the reviewer that saving time is not the main reason for using the expert elicitation methodology. On the other hand, depending on the content and complexity of the study, elicitation could potentially provide results in a shorter time.

Considering our study, setting up and individually calibrating a single hydrological model for a specific catchment could easily take up one year of work for an individual. Even though several scientists assisted in the preparation for the elicitation workshop, overall it did not take the same amount of working hours to design the elicitation procedure, prepare the training

**material and contact the experts. Even though the time required to plan and perform the elicitation is comparatively shorter than the time required to, for instance setup and calibrate models, the main goal of an elicitation is to consider expert knowledge for informed decision-making.**

Line 476: "Climate models often stem from short-term forecast models". Taken over many decades of model development this is true. However, this is a bit misleading as CMIP5 climate models are quite different from operational numerical weather prediction models. Suggest clarifying what is meant here – yes, climate change is more of a boundary value problem but scientists don't simply add on elements to a NWP model to simulate future climate – there is a quite a lot more involved.

**We agree with the reviewer, and have revised the sentence to reflect that there is more to the models than just the time scale that they are applied to simulate:**

"Climate models often stem from short-term forecast models, which have evolved over decades as part of the process of changing emphasis from an initial value problem to a boundary condition problem, which also shifts focus from calibration to the predicted surface variables to the main energy and water budget and main climate processes (Hourdin et al., 2017)." (lines 509-512)

Line 522: Sentence ending "...are without doubt inappropriate". This phrasing is too strong given the evidence. Had the climate modellers been provided with compelling evaluation information, I'm sure they would have been open to excluding models. Suggest deleting "without doubt" and rephrasing.

We agree with the reviewer that in other expert elicitation studies with different information and a different elicitation setup climate modellers could in fact exclude or down-rank climate models. This is included in the manuscript at lines 552-555: 'We believe that with a different design for the expert elicitation that provides climate experts with more information on hydrological impact simulations and aims at clarifying how to best select a few RCM projections that best sample the spread of climate projections, the elicitation could potentially have resulted in down-ranking some model combinations of the ensemble.'

We deleted the end of the sentence not to make such a strong statement: '*The conclusion of our* study is that in this expert elicitation setting climate modellers deem each climate model to have equal probability and are very unlikely to exclude or down-rank any climate model.' lines (555-556)

Further thoughts for the discussion section:

1. It would be useful to comment further on the uncertainty cascade, referenced in the introduction section. In particular, does having expert opinion included in the articulation of uncertainties add yet another layer of cascading uncertainties? Or does it rather try to address and reduce the cascading uncertainties? It isn't obvious to me.

The intention of this expert elicitation methodology was to assess probabilities within an ensemble. In our opinion the experts here work a bit outside the uncertainty cascade not adding to it, but not reducing it either, but rather reflecting on it. A sentence was added to the manuscript: 'In our opinion expert elicitation does not add an extra layer of uncertainty to the uncertainty cascade of hydrological impact modelling. As experts tend to focus on large uncertainties that are not easily quantifiable by direct metrics, they rather point out uncertainties that are not necessarily obvious thereby increasing our knowledge about uncertainties and assisting decision making.' lines (437-440)

In general, climate models (GCMs and RCMs) are more complicated and have higher dimensionality than hydrological models. Could this be a reason why climate modellers prefer model democracy, especially if they aren't entirely familiar with all aspects of the models?

The reviewer gave a good summary of why the climate modellers were more likely to prefer model democracy. We included a sentence about the complexity of climate models in the section where the differences between the model communities are discussed. *Climate models in general are also more complex and have higher dimensionality than hydrological models, thus climate modellers are less likely to be familiar with all aspects of a climate model which could in turn significantly affect climate impacts.* (lines 500-502)

2. (with particular reference to lines 444 to 448) Another reason why experts won't assign different weights to the climate models may be because they are all from the same generation – i.e., all RCMs downscaled with CMIP5 models. Might the result be different if comparing CMIP3 vs CMIP6 models for example?

We agree with the reviewer that using climate models from different generations would most likely result in different weights or at least in some kind of down-ranking instead of using the uniform weighting as seen in the study. This would probably stem from the different resolution, model performance, included processes etc. of the model generations.

However as our study aimed at using the most up-to-date climate models at the time of the elicitation, we do not intend to address this issue in our manuscript, opening up a new discussion about the differences in generations of climate models and how these differences could potentially influence the elicitation methodology.

3. Sample size is an issue for this study. With only 6 experts in each group, any result cannot be considered robust - i.e., the finding that 6 hydrological modellers were more willing to assign weights compared to 6 climate modellers, is not a robust finding. It would be good to comment on sample size limitations.

We agree with the reviewer that 6 experts may not provide the most reliable results. On the other hand we would like to point out that the elicitation setup of this study may not allow for many more experts, originally as reaching consensus becomes more time consuming with the increasing number of experts present and later as being forced to a virtual environment, reaching consensus is even lengthier process than in a personal workshop. Using only individual elicitation, there is no limitation on the number of experts involved in a study. There is however an upper limit where involving further experts does not contribute with further information, (Cooke and Probst, 2006). Using individual elicitation only, the discussion part of the elicitation is missing. As concluded in our manuscript, this final consensus stage was really necessary, especially for the climate modeller group who finally assigned equal probability to all climate models in the last consensus step of the elicitation.

We added a sentence about the number of experts in the manuscript.:' The two virtual workshops were planned with the participation of these 12 experts; 6 hydrology and 6 climate experts corresponding to Cooke and Probst, 2006 who specified 6 experts as the minimum number to obtain robust results and 12 as an upper limit where additional experts do not contribute with further benefits.' (lines 194-197)

4. At the end of the conclusions section, you comment on the impact of covid-19 moving to virtual engagement. I suggest moving some of this to the discussion section and elaborating more on the methodological implications and insights that may be relevant to other studies in the future.

**Please see our response to comment on line 177.**

All the best in revising the paper and I look forward to seeing the published article.

**Use of expert elicitation to assign weights to climate and hydrological models in climate impact studies**

Eva Sebok1, Hans Jørgen Henriksen1, Ernesto Pastén-Zapata1,2, Peter Berg3, Guillaume Thirel4, Anthony Lemoine4, Andrea Lira-Loarca5, Christiana Photiadou3,6, Rafael Pimentel7,8, Paul Royer-Gaspard4, Erik Kjellström3, Jens Hesselbjerg Christensen9,10,11, Jean Philippe Vidal12, Philippe Lucas-5 Picher13,14, Markus G. Donat15,16, Giovanni Besio17, María José Polo7,8, Simon Stisen1, Yvan Caballero 18, Ilias G. Pechlivanidis3, Lars Troldborg1, Jens Christian Refsgaard1 1Geological Survey of Denmark and Greenland, Copenhagen, Denmark 2University of Eastern Finland, Joensuu, Finland 3Swedish Meteorological and Hydrological Institute, Norrköping, Sweden 10 4Université Paris-Saclay, INRAE, HYCAR Research Unit, Antony, France 5University of Granada, Granada, Spain 6European Environment Agency, Copenhagen, Denmark 7Fluvial Dynamics and Hydrology Research Group, Andalusian Institute for Earth System Research (IISTA), University of Córdoba, Córdoba, Spain 15 8Department of Agronomy, Unit of Excellence María de Maeztu (DAUCO), University of Córdoba, Córdoba, Spain 9Physics of Ice, Climate and Earth, Niels Bohr Institute, University of Copenhagen, Copenhagen, Denmark 10NORCE Norwegian Research Centre, The Bjerknes Centre for Climate Research, Bergen, Norway 11Danish Meteorological Institute, Copenhagen, Denmark 20 12INRAE, UR RiverLy, Villeurbanne Cedex, France 13Groupe de Météorologie de Grande Échelle et Climat, Centre National de Recherches Météorologiques, Météo-France, Toulouse, France 14Département des sciences de la Terre et de l'atmosphère, Université du Québec à Montréal, Montréal, Québec, Canada 15Barcelona Supercomputing Center, Barcelona, Spain 25 16ICREA, Pg. Lluís Companys 23, Barcelona, Spain 17University of Genoa, Genoa, Italy 18BRGM, 
[revised manuscript text omitted]
               | Swadan         | Dommonly       | Eronaa         | Guadalfeo,     | Granada,       |
|--------------------------|----------------|----------------|----------------|----------------|----------------|
| location                 | Sweden         | Denmark        | France         | Spain          | Spain          |
|                          |                |                | Climate        | Climate        | Changes in     |
|                          |                | Climate        | change         | change         | physical       |
|                          | Climate        | change         | impacts on     | impacts on the | processes such |
| Aim of the               | change         | impacts on the | hydropower     | allocation of  | as sea waves,  |
| assessment               | impacts on the | foundation of  | production     | water for      | fluvial        |
|                          | biodiversity   | agricultural   | and water      | tourism,       | discharges and |
|                          |                | production     | management     | agriculture    | sediment       |
|                          |                |                | for other uses | and energy     | transport      |
| Area (km 2 )  | 10,000         | 1,124          | 3,580          | 530,5          |                |
| Precipitation
(mm/yr) | 741            | 1,003          | 1,055          | 745            | 460 - 630      |
| Temperature
(Celsius) | 5.6            | 8.8            | 3.2            | 12.5           | -              |
| Wind speed
(m/s)      | -              | -              | -              | -              | 18-22          |
| Wind waves
(km)       | -              | _              | -              | _              | 200 - 300      |

Table 1. Aim of the different study cases along with their physical and observed climate characteristics

| GCM        | RCM        | Realization (Member #) |
|------------|------------|------------------------|
| EC-EARTH   | RACMO22E   | 12                     |
| EC-EARTH   | CCLM4-8-17 | 12                     |
| EC-EARTH   | RCA4       | 12                     |
| HadGEM2-ES | RCA4       | 1                      |
| HadGEM2-ES | RACMO22E   | 1                      |
| MPI-ESM-LR | RCA4       | 1                      |
| MPI-ESM-LR | REMO2009   | 1                      |

Table 2 Climate model ensemble used for expert elicitation. The realization (member #) column denotes the version of initial conditions.

| Site                          | Models                 | Characteristics                                                                                                                                                            | References              |
|-------------------------------|------------------------|----------------------------------------------------------------------------------------------------------------------------------------------------------------------------|-------------------------|
| Denmark                       | MIKE-SHE:
Two-layer | • Physically-based, integrated and fully distributed model at a resolution of 250m x                                                                                       | Abbott et al., 1986;    |
|                               |                        | 250m                                                                                                                                                                       | Graham and Butts,       |
|                               |                        | • The model divides the unsaturated zone into a root zone and a zone below the root zone                                                                                   | 2005; Yan and Smith,    |
|                               |                        | • Vertical flow depends on the storage of the unsaturated zone, ignoring the delay                                                                                         | 1994                    |
|                               |                        | in the flow                                                                                                                                                                |                         |
|                               |                        | • Actual evaporalispitation is a function of the potential evaporalispitation and the soil moisture content                                                                |                         |
|                               | MIKE-SHE               | • Physically-based, integrated and fully distributed model at a resolution of 250m x                                                                                       | Abbott et al., 1986;    |
|                               | Gravity                |  <li>250m</li> <li>Based on the continuity equation and Darcy's law</li>                                                                                          | Graham and Butts,       |
|                               | flow                   |  <li>Vertical flow only depends on the force of gravity</li>                                                                                                      | 2005; Kristensen and    |
|                               | now                    | • Actual evapotranspiration is a function of vegetation and the soil moisture content                                                                                      | Jensen, 1975            |
|                               | MIKE OHE               | • Physically-based, integrated and fully distributed model at a resolution of 250m x                                                                                       | Abbott et al., 1986;    |
|                               | Dichards'              | 250m                                                                                                                                                                       | Graham and Butts,       |
|                               | Richards               |  <li>Based on the continuity equation and Darcy's law</li> <li>Vertical flow depends on the soil moisture retention and hydraulic conductivity</li>               | 2005; Kristensen and    |
|                               | equation               |  <li>Actual evapotranspiration is a function of vegetation and the soil moisture content</li>                                                                     | Jensen, 1975            |
|                               | GR4J                   | Lumped conceptual model with four parameters                                                                                                                               | Perrin et al., 2003;    |
|                               |                        | • Water balance controlled by actual evapotranspiration and groundwater                                                                                                    | Valéry et al., 2014     |
|                               |                        | Snowmelt is simulated using the two-parameter CemaNeige model                                                                                                              | Perrin 2000: Le         |
|                               | GR6J                   |  <li>Modified version of GR4J that allows a change of the direction of the exchange</li>                                                                          | Moine 2008: Valéry      |
| Franco                        |                        | and adds a conceptual store                                                                                                                                                | et al 2014              |
| Trance                        |                        | Snowmelt is simulated using the two-parameter CemaNeige model                                                                                                              | Mighal at al 2002:      |
|                               | ТОРМО                  |  <li>Lumped conceptual model adapted from IOPMODEL with seven parameters</li> <li>Water balance controlled by evaporation from interception and groundwater</li>  | December of all, 2005,  |
|                               |                        | reservoirs                                                                                                                                                                 | beven and Kirvy,        |
|                               |                        | • Simulates runoff combining heterogeneous time response flows from exponential                                                                                            | 1979; Valery et al.,    |
|                               |                        |  <li>Snowmelt is simulated using the two-parameter CemaNeige model</li>                                                                                           | 2014                    |
|                               | НУРЕ                   | Semi-distributed conceptual model                                                                                                                                          | Lindström et al., 2010; |
| Spain
(Guadalfeo
River) |                        | • Infiltration is estimated using a water table discrimination model                                                                                                       | Samuelsson et al.,      |
|                               |                        | • Evaporation is estimated using the modified Hargreaves-Samani method                                                                                                     | 2011                    |
|                               |                        | • Snowmelt is simulated using three decay factors related to temperature, radiation and fractional snow cover                                                              |                         |
|                               | SWAT                   | Semi-distributed conceptual model                                                                                                                                          | Arnold et al., 1988;    |
|                               |                        | • Infiltration is estimated using the Green and Ampt method for a single soil layer                                                                                        | Green and Ampt,1911;    |
|                               |                        | • Evapotranspiration is estimated using the Penman-Monteith method                                                                                                         | Penman 1948;            |
|                               |                        | Showmen is simulated using the degree-day method                                                                                                                           | Monteith et al., 1964   |
|                               | WiMMed                 | Distributed physically-based model                                                                                                                                         | Polo et al., 2009;      |
|                               |                        | • Infiltration is estimated using the Green and Ampt method for a two-soil layer                                                                                           | Aguilar et al., 2011,   |
|                               |                        | Evapotranspiration is estimated using the Penman-Monteith method                                                                                                           | Herrero et al., 2009,   |
|                               |                        | Showmen is simulated using a punctual energy and mass balance extended to cell scale using depletion curves                                                                | Pimentel et al., 2017,  |
| L                             | l                      |                                                                                                                                                                            | l                       |

Table 3 Hydrological models used in each of the study sites

| Danish case           | Step 1 50th | Step 2 50th | Group consensus |  |
|-----------------------|-------------|-------------|-----------------|--|
| study                 | percentile  | percentile  | Group consensus |  |
| Two-Layer             | 0.4         | 0.38        | 0.38            |  |
| Gravity flow          | 0.2         | 0.25        | 0.27            |  |
| Richards'
equation | 0.35        | 0.35        | 0.35            |  |
| French case           | Step 1 50th | Step 2 50th | Group consensus |  |
| study                 | percentile  | percentile  |                 |  |
| GR4J                  | 0.35        | 0.33        | 0.35            |  |
| GR6J                  | 0.33        | 0.33        | 0.35            |  |
| ТОРМО                 | 0.30        | 0.33        | 0.3             |  |
| Spanish case          | Step 1 50th | Step 2 50th | Group consensus |  |
| study                 | percentile  | percentile  |                 |  |
| НҮРЕ                  | 0.18        | 0.35        | 0.25            |  |
| SWAT                  | 0.45        | 0.37        | 0.30            |  |
| WiMMed                | 0.42        | 0.40        | 0.45            |  |

835 Table 4 Assessed probabilities by hydrological model experts

| Danish case study       | Step 1 50th percentile | Step 2 50th percentile | Group consensus |
|-------------------------|------------------------|------------------------|-----------------|
| 1-EC-EARTH-CCLM         | 0.21                   | 0.16                   | 0.14            |
| 2-EC-EARTH-RACMO        | 0.13                   | 0.19                   | 0.14            |
| 3-EC-EARTH-RCA4         | 0.09                   | 0.08                   | 0.14            |
| 4-HadGEM-RACMO          | 0.18                   | 0.17                   | 0.14            |
| 5-HadGEM-RCA4           | 0.14                   | 0.14                   | 0.14            |
| 6-MPI-ESM RCA4          | 0.09                   | 0.08                   | 0.14            |
| 7-MPI-ESM REMO          | 0.14                   | 0.14                   | 0.14            |
| French case study       | Step 1 50th percentile | Step 2 50th percentile | Group consensus |
| 1-EC-EARTH-CCLM         | 0.20                   | 0.18                   | 0.14            |
| 2-EC-EARTH-RACMO        | 0.12                   | 0.10                   | 0.14            |
| 3-EC-EARTH-RCA4         | 0.11                   | 0.13                   | 0.14            |
| 4-HadGEM-RACMO          | 0.15                   | 0.15                   | 0.14            |
| 5-HadGEM-RCA4           | 0.12                   | 0.14                   | 0.14            |
| 6-MPI-ESM RCA4          | 0.10                   | 0.12                   | 0.14            |
| 7-MPI-ESM REMO          | 0.15                   | 0.18                   | 0.14            |
| Spanish case study 1 SN | Step 1 50th percentile | Step 2 50th percentile | Group consensus |
| 1-EC-EARTH-CCLM         | 0.14                   | 0.16                   | 0.14            |
| 2-EC-EARTH-RACMO        | 0.13                   | 0.15                   | 0.14            |
| 3-EC-EARTH-RCA4         | 0.12                   | 0.14                   | 0.14            |
| 4-HadGEM-RACMO          | 0.14                   | 0.14                   | 0.14            |
| 5-HadGEM-RCA4           | 0.13                   | 0.14                   | 0.14            |
| 6-MPI-ESM RCA4          | 0.08                   | 0.08                   | 0.14            |
| 7-MPI-ESM REMO          | 0.09                   | 0.09                   | 0.14            |
| Spanish case study 2 G  | Step 1 50th percentile | Step 2 50th percentile | Group consensus |
| 1-EC-EARTH-CCLM         | 0.14                   | 0.16                   | 0.14            |
| 2-EC-EARTH-RACMO        | 0.15                   | 0.16                   | 0.14            |
| 3-EC-EARTH-RCA4         | 0.11                   | 0.10                   | 0.14            |
| 4-HadGEM-RACMO          | 0.15                   | 0.15                   | 0.14            |
| 5-HadGEM-RCA4           | 0.14                   | 0.10                   | 0.14            |
| 6-MPI-ESM RCA4          | 0.09                   | 0.09                   | 0.14            |
| 7-MPI-ESM REMO          | 0.11                   | 0.12                   | 0.14            |
| Swedish case study      | Step 1 50th percentile | Step 2 50th percentile | Group consensus |

| 1-EC-EARTH-CCLM  | 0.14 | 0.18 | 0.14 |
|------------------|------|------|------|
| 2-EC-EARTH-RACMO | 0.14 | 0.15 | 0.14 |
| 3-EC-EARTH-RCA4  | 0.10 | 0.09 | 0.14 |
| 4-HadGEM-RACMO   | 0.18 | 0.19 | 0.14 |
| 5-HadGEM-RCA4    | 0.14 | 0.16 | 0.14 |
| 6-MPI-ESM RCA4   | 0.11 | 0.12 | 0.14 |
| 7-MPI-ESM REMO   | 0.14 | 0.15 | 0.14 |
|                  |      |      |      |

Table 5 Assessed probabilities by climate model experts

---

## Author Response (AR2)

**Response to the editor's comment to the manuscript: '***Use of expert elicitation to assign weights to climate and hydrological models in climate impact studies***' (manuscript number hess-2021-597)**

Thank you for carefully considering the remarks and suggestions of two anonymous reviewers that provided excellent comments to your article and brought extremely important issues in the application of climate model projections for climate impact studies. I have very much enjoyed reading your responses to the reviewers. Your clarifications and insights on both reviewer's' concerns about the expert elicitation methodology you are proposing, provides a good degree of transparency on your scientific contributions, adding an extra layer of clarity in this important effort. My only comment refers to the use of the wording "assign equal probability to all climate models" or using the wording "equal probability". I understand this refer to "an equal probability for all models to reliably projecting future climate changes", but reading the statement by itself can be misleading. I would clarify this statement by adding one two sentences at beginning of the methodology about what the word "probability" means in this context of expert elicitation. You also use the word "weights" as a synonymous of probability, but this also deserves clarification.

**A section explaining the terms 'equal probability' and 'weight' was added to the methods section: '***In the context of this study equal probability means that all models have an equal probability of correctly predicting future climate change or the impacts of climate change on hydrology in case of hydrological models. While describing the results of the study, the term 'model weight' is used interchangeably with 'model probability*****'. (lines 221-224)**